# Serum Amyloid A is not obligatory for high-fat, high-sucrose, cholesterol-fed diet-induced obesity and its metabolic and inflammatory complications

Ailing Ji[1,2], Andrea C. Trumbauer[2], Victoria P. Noffsinger[1,2], Hayce Jeon[3,4], Avery C. Patrick[3], Frederick C. De Beer[1,2,5], Nancy R. Webb[3,2,5], Lisa R. Tannock[1,2,5,6], Preetha Shridas[1,2,5]*

1 Department of Internal Medicine, University of Kentucky, Lexington, Kentucky, United States of America, 2 Saha Cardiovascular Research Center, University of Kentucky, Lexington, Kentucky, United States of America, 3 Pharmacology and Nutritional Sciences, University of Kentucky, Lexington, Kentucky, United States of America, 4 Department of Obstetrics and Gynecology, University of Kentucky, Lexington, Kentucky, United States of America, 5 Barnstable Brown Diabetes Center, University of Kentucky, Lexington, Kentucky, United States of America, 6 Department of Veterans Affairs (LRT), Lexington, Kentucky, United States of America

* pshri2@uky.edu

**Data Availability Statement:** All relevant data are within the paper and its Supporting Information files.

## Abstract

Several studies in the past have reported positive correlations between circulating Serum amyloid A (SAA) levels and obesity. However, based on limited number of studies involving appropriate mouse models, the role of SAA in the development of obesity and obesity-related metabolic consequences has not been established. Accordingly, herein, we have examined the role of SAA in the development of obesity and its associated metabolic complications *in vivo* using mice deficient for all three inducible forms of SAA: SAA1.1, SAA2.1 and SAA3 (TKO). Male and female mice were rendered obese by feeding a high fat, high sucrose diet with added cholesterol (HFHSC) and control mice were fed rodent chow diet. Here, we show that the deletion of SAA does not affect diet-induced obesity, hepatic lipid metabolism or adipose tissue inflammation. However, there was a modest effect on glucose metabolism. The results of this study confirm previous findings that SAA levels are elevated in adipose tissues as well as in the circulation in diet-induced obese mice. However, the three acute phase SAAs do not play a causative role in the development of obesity or obesity-associated adipose tissue inflammation and dyslipidemia.

## Introduction

Obesity is associated with a wide variety of pathologies, collectively known as metabolic diseases resulting in significant morbidity and mortality. Chronic and low-grade inflammation is a hallmark of obesity and key factor for the development of obesity comorbidities [1]. Obesity-driven adipose tissue inflammation has been extensively characterized and its role in the

**Funding:** This study was funded by the Diabetes Research Center at Washington University in St. Louis of the National Institutes of Health in the form of a grant (No. P30DK020579) to PS. This study was also funded by the National Institutes of Health Grants in the form of grants to PS and LT (No. HL147381), and NW and FD (No. HL134731). This study was also funded by the Department of Veterans Affairs in the form of a grant to LT (No. BX004275). This study was also supported with resources and facilities provided by the Centers of Biomedical Research Excellence (COBRE) at the University of Kentucky, which was supported by an Institutional Development Award (IDeA) from the National Institute of General Medical Sciences of the National Institutes of Health under a grant (No. P30 GM127211). The funders had no role in study design, data collection, and analysis, decision to publish, or preparation of the manuscript.

**Competing interests:** The authors have declared that no competing interests exist.

development of insulin resistance and impaired glucose metabolism leading to the development of type 2 diabetes have been established [2, 3]. However, the obesity-induced factors responsible for the inflammatory state in adipose tissue and the relationship of dysregulated adipose tissue to systemic inflammation still remain unclear.

Serum amyloid A (SAA) is a family of acute phase proteins whose circulating levels rise (up to 1,000-fold or more) in an acute phase response. There are three inducible SAA subtypes that likely arose through gene duplication [4]. Humans express two acute-phase SAA proteins, SAA1 and SAA2 which are 96% homologous over their entire length, and correspond to mouse SAA1.1 and SAA2.1. In addition, mice encode a third conserved acute-phase SAA gene, *Saa3*. *Saa3* is considered to be a pseudogene in humans due to an early stop codon [5–7]. The liver is believed to be the predominant source of circulating SAA during an acute inflammatory state and in chronic inflammatory diseases such as rheumatoid arthritis and lupus, although we [8] and others [9, 10] have shown that all three SAA isoforms are also significantly induced in adipose tissue in endotoxemic mice. Yet another isoform of SAA, SAA4, is expressed in humans and mice and is synthesized constitutively [11]. Unlike the three acute-phase isoforms, SAA4 does not show massive induction during acute inflammation [7, 11]. Chronic and modest elevations in SAA concentrations have been demonstrated in obesity, metabolic syndrome or diabetes [12–14], and circulating concentrations of SAA correlate with body fat [15]. Weight loss tends to decrease these levels [15–18]. In a prospective population-based study, a statistically significant association between systemic SAA levels and the development of type 2 diabetes was observed in an elderly western European population that was independent of various other established type 2 diabetes risk factors [19]. However, though SAA levels are positively correlated with both obesity and diabetes, whether there is any physiological relevance to this association is not clear. Several *in vitro* and selected *in vivo* studies demonstrated SAA to invoke inflammatory properties and functions that would be expected to promote the development of obesity, inflammation and insulin resistance. Studies with recombinant SAA have indicated that SAA acts as a chemoattractant for both monocytes and polymorphonuclear cells [20]. When not associated with high density lipoprotein (HDL), SAA is known to increase the production of cytokines, reactive oxygen species and nitric oxide [21–24]. The results from *in vitro* studies using recombinant SAA are questionable now as differences between recombinant SAA and endogenous SAA purified from acute-phase plasma have been found [25]. Our studies using SAA purified from mouse plasma shows activation of NLRP3 inflammasome activation in mouse macrophage cells [26]. A previous report indicated that SAA induced hypoxia, a common event associated with fat expansion [27].

Earlier studies in mice indicated that deficiency of SAA3 blunts weight gain induced by obesogenic diet, hyperlipidemia and adipose tissue specific inflammation and macrophage accumulation are attenuated in female but not in male mice [28]. Ahlin et al. studied the role of human SAA1 overexpression in adipose tissues of mice, and observed no significant impact on diet-induced obesity, adipose tissue inflammation and insulin resistance [29]. A study by de Oliveira et al. [30] showed that suppression of SAA in male Swiss Webster mice by SAA-1 and 2 -targeted antisense oligonucleotides caused a significant reduction in adipose tissue expansion, expression of inflammatory markers, macrophage infiltration into the adipose tissues and remarkable improvement in glucose and insulin tolerance in mice fed a high fat diet. It is not yet clear whether the SAA isoforms compensate for the deficiency of each other and whether deficiency of all SAA subtypes impacts obesity-associated inflammation and metabolic dysfunction. Although it is now widely accepted that increased adiposity leads to elevated SAA in adipose tissue and serum in both humans and mice, evidence that SAA plays a key role in obesity-associated metabolic dysfunction is somewhat lacking. Whether SAA plays a functional role in the development of obesity and/or obesity-induced adipose tissue inflammation,

insulin resistance and other metabolic complications, or is merely a consequence of adipose tissue inflammation and thus a marker of obesity remains an unresolved question. SAA is involved in the development of several chronic inflammatory diseases, including atherosclerosis [31, 32] and angiotensin II-induced abdominal aortic aneurysm formation [33]. Here, we investigated whether SAA plays a causal role in the development of diet-induced obesity and associated metabolic changes in mice.

## Materials and methods

### Animals

C57BL/6 mice deficient in SAA1.1, SAA2.1 and SAA3 (TKO) mice were generously provided by Drs. June-Yong Lee and Dan Littman, New York University. The TKO mice were generated by inserting a premature stop codon into exon 2 of *saa3* in the SAA1.1/2.1-deficient (SAAKO) mouse [33, 34] using CRISPR-Cas9 technology as described previously [35, 36]. The details of the genomic organization of the *SAA1.1* and *SAA2.1* genes and the construction of SAAKO mice were described earlier [32]. Genotyping to identify SAA TKO mice is performed using a multi-step approach. The first step confirmed targeting of the *SAA1.1* and *SAA2.1* loci utilizing a 3-primer PCR reaction as described earlier [37]). The second step to validate *SAA3* targeting is performed by two sets of PCRs as described in S1A Fig. All mice were provided food and water *ad libitum*. Animals were housed in micro-isolator cages and maintained on a 14-hr light/10-hr dark cycle. All studies were performed in accordance with the Public Health Service Policy on Humane Care and Use of Laboratory Animals with the approval of the University of Kentucky Institutional Animal Care and Use Committees.

### Diet and study design

The study design is depicted in Fig 1. Briefly, 17-weeks old male and female C57BL/6 (WT) and TKO mice were fed either normal chow (18% calories from fat, 58% from carbohydrate and 24% from protein) or a high fat, high-sucrose diet with 0.15% added cholesterol (HFHSC, F4997, Bioserv, Flemington, NJ) *ad libitum* for 16 weeks. The diet regimen was selected based on previous studies, where it significantly induced SAA expression in adipose tissues and plasma, and increased macrophage accumulation in adipose tissues [28, 38]. The HFHSC diet

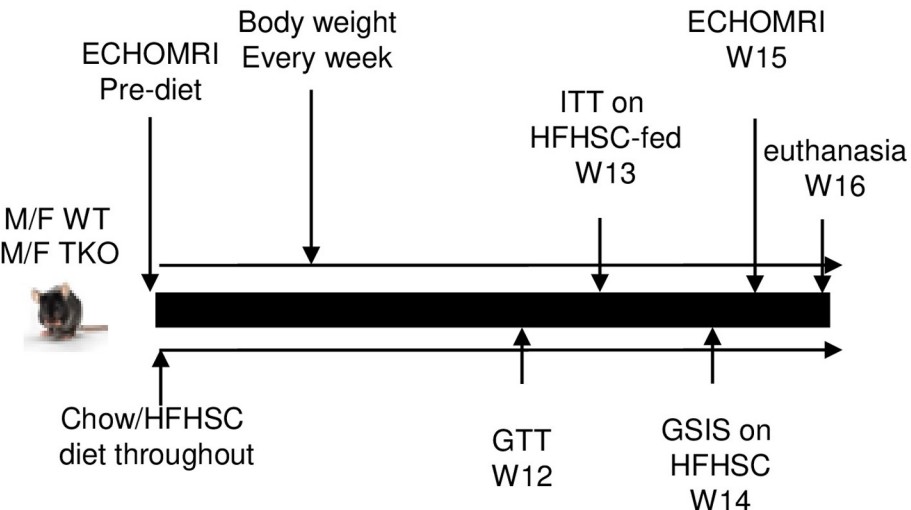

**Fig 1. Study design.**

provides 20.5% of calories as protein, 36% as fat and 36% as carbohydrate. Body weight was measured weekly and body composition was measured by NMR spectroscopy (Echo MRI).

### Intraperitoneal glucose tolerance test (GTT)

Mice were fasted for 4 h and then given intraperitoneal (*i.p.*) injection of D-(+)-glucose (Sigma-Aldrich, St. Louis, MO) at 1.5 g/kg body weight. Blood glucose was measured in tail vein blood (Contour Blood glucose monitoring system) just prior to injection of glucose (time 0) and at 30, 60, 90, 120, 150 and 180 min after glucose administration.

### Intraperitoneal insulin tolerance test (ITT)

Mice were fasted for 4 h, and then given human insulin (Novolin R) *i.p.* at 1.0 IU/kg body wt. Blood glucose was measured at 0, 30, 60, 90, 120, and 150 min post injection from tail vein prick. None of the mice developed hypoglycemic shock (glucose levels below 20 mg/dL) and unresponsive to touch as a result of hypoglycemia.

### Glucose-stimulated insulin secretion

Glucose-stimulated insulin secretion (GSIS) was performed as described earlier [39]. Briefly, mice were fasted for 16 h, anesthetized, and then blood samples were collected from the retro-orbital sinus before and 15 min after intraperitoneal glucose injection (3 g/kg). Insulin levels in the plasma samples were determined.

### Plasma SAA measurements

Plasma SAA concentrations were determined using a mouse SAA ELISA kit (Tridelta Development Ltd, Maynooth, County Kildare, Ireland). Plasma SAA3 concentrations were determined using a mouse SAA3 ELISA kit (Millipore Sigma, Burlington, MA).

### Plasma lipid analyses

Plasma total cholesterol, free fatty acid, triglyceride and HDL concentrations were measured using enzymatic kits (Wako Chemicals, Richmond VA.).

### Liver triglyceride and cholesterol quantitation

Triglycerides from liver samples were measured according to a previously published method [40]. Briefly, ~50 mg of liver tissue was homogenized in 1 ml of chloroform-methanol (2:1 v/v). The homogenate was then diluted 1:10 with chloroform-methanol (2:1 v/v) and 10 μl of the diluted homogenate was evaporated. The evaporated sample was resuspended directly in 200 μl of Triglycerides Reagent (Pointe Scientific, Canton, MI) and the assay was performed following the manufacture's description. Cholesterol assay was performed following Folch method [41] with modifications. Briefly, similar to triglyceride assay, ~50 mg of liver tissue was homogenized in 1 ml of chloroform-methanol (2:1 v/v). The homogenate (50 μl) was evaporated and resuspended in 1% Triton X-100 in water. Aliquots were assayed for cholesterol content using a colorimetric kit (Wako Chemicals, Richmond VA).

### RNA isolation and quantitative RT-PCR

Total RNA was isolated from mouse adipose tissues according to the manufacturer's instructions (Qiagen, Valencia, CA). RNA samples were incubated with DNase I (Qiagen, Valencia, CA) for 15 min at RT prior to reverse transcription. Adipose tissue RNA (0.25–0.5 μg) was

reverse transcribed into cDNA using the Reverse Transcription System (Applied Biosystems, Waltham, MA). After 4-fold dilution, 5 μl was used as a template for real-time RT-PCR. Amplification was done for 40 cycles using Power SYBR Green PCR master Mix Kit (Applied Biosystems, Waltham, MA). Quantification of mRNA was performed using the ΔΔCT method and normalized to GAPDH. The primers used for the quantification of SAA mRNA were designed to recognize all three inducible isoforms of mouse SAA, SAA1.1, SAA2.1 and SAA3 (Forward: `GACATGTGGCGAGCCTAC` and reverse: `TTGGGGTCTTTGCCACT`). In some studies, SAA3 mRNA abundance was specifically quantified using primers specific for SAA3 mRNA (Forward: `TTTCTCTTCCTGTTGTTCCAGTC` and reverse: `TCCCAATGTGCTGAATAAATAC` `TTGTGA`). Other primer sequences will be provided on request.

## Western blotting

Adipose tissues from the experimental mice were homogenized in RIPA buffer (Sigma-Aldrich, St. Louis, MO) containing a protease inhibitor cocktail (Sigma-Aldrich, St. Louis, MO). The homogenate (50 μg protein) was electrophoresed on a 4–50% polyacrylamide gradient gel (Bio-Rad, Hercules, CA) and immunoblotted with a mixture of two antibodies (rabbit anti-mouse SAA1 + SAA2 antibody, 1: 1000 dilution, Cat number ab199030; abcam, Cambridge, UK) and rabbit anti-mouse SAA3 (a gift from Dr. Phillip Scherer, University of Texas Southwestern) to identify all three acute-phase SAA isoforms, SAA1.1, SAA2.1 and SAA3. The specificity of each of these antibodies has been validated in a previous study [8].

## Histology

Gonadal fat tissues were collected from the experimental mice and fixed in 10% formaldehyde, paraffin embedded, cut into 5-μm sections, and stained with hematoxylin (Vector laboratories, Burlingame, CA). Adipocyte area was quantified for 3 randomly selected sections from 3 mice from each group of experimental mice using Nikon NIS-elements software.

## Statistics

Data are expressed as mean ±SEM. Results were analyzed by Student's *t* test or one-way analysis of variance followed by Sidak's multiple comparison test. Values of $p < 0.05$ were considered statistically significant.

# Results

## SAA expression is induced with HFHSC diet feeding

To investigate whether SAA plays a role in the development of diet-induced obesity and insulin resistance, male and female wild type (WT) and SAA-deficient (TKO) mice were fed either normal rodent diet or a high fat, high sucrose diet with added cholesterol (HFHSC) *ad libitum* for 16 weeks (study design in Fig 1). At study termination, total SAA mRNA abundance in gonadal adipose tissue was significantly increased in both male (9-fold increase) and female (1.6-fold increase) WT mice fed HFHSC diet compared to the chow fed mice of the same sex (Fig 2A). SAA protein was detected by immunoblotting in adipose tissue of WT mice fed HFHSC, but not chow diet (Fig 2B). As expected, SAA protein was not detected in the adipose tissues of SAA-TKO mice fed either chow or HFHSC diets. There was a trend for increased SAA mRNA expression in the livers of WT mice fed HFHSC diet compared to chow, however the difference was not significant in either male or female mice (Fig 2C). As expected, SAA mRNA was undetectable in livers of TKO mice (Fig 2C).

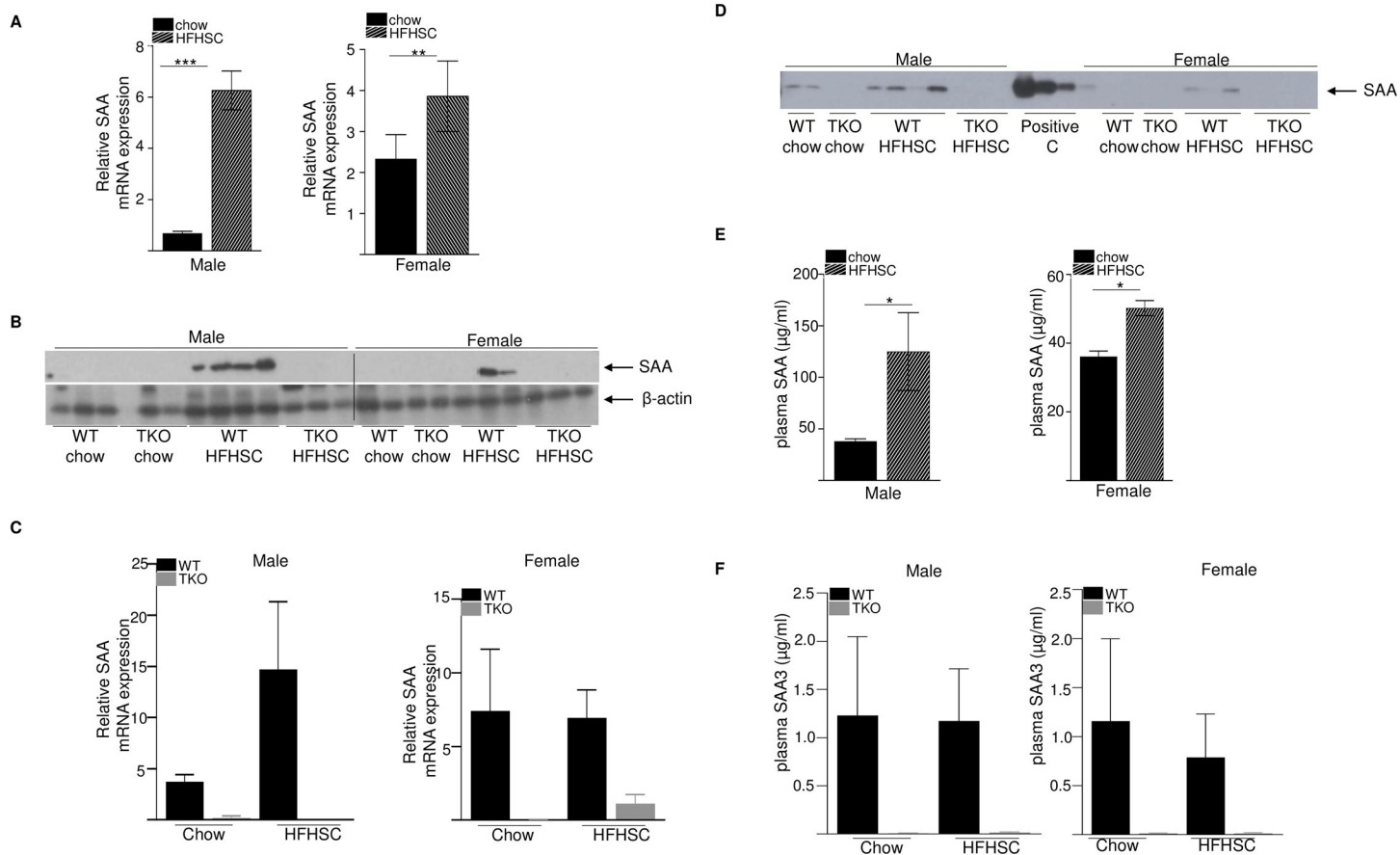

**Fig 2. SAA expression is significantly induced in the adipose tissues of obese mice.** A) SAA mRNA abundance in gonadal adipose tissue from chow and HFHSC fed WT male (left panel) and female (right panel) mice after 16 weeks of diet feeding (n = 5 mice/group) B) Adipose tissue lysates (50 μg protein/lane) obtained from WT and TKO mice at the end of the study were immunoblotted for SAA (top panel) and β-actin as loading control (bottom panel). SAA proteins were undetectable in the adipose tissues of chow fed WT mice and all TKO mice. C) SAA mRNA abundance in liver tissue from chow and HFHSC fed WT and TKO male (left panel) and female (right panel) mice after 16 weeks of diet feeding (n = 4–5 mice/group). D) Plasma SAA in WT and TKO mice at the end of the study by western blot. 1 μl plasma was used/lane. E) SAA levels by elisa in chow and HFHSC fed WT male and female mice after 16 weeks of diet feeding. F) plasma SAA3 levels by elisa in chow and HFHSC fed WT male and female mice after 16 weeks of diet feeding. Data are mean ±SEM; * = P ≤ 0.05, ** = P ≤ 0.01 and *** = P ≤ 0.001.

SAA was detected by western blot in the plasmas of WT male mice fed either chow or HFHSC diet, however in WT female mice, SAA was barely detectable only in the HFHSC fed mice (Fig 2D). As expected, SAA was undetectable in both chow or HFHSC diet fed male or female TKO mice (Fig 2D). Consistent with the western blot analysis, analysis of plasma SAA (SAA1.1 and SAA2.1, the predominant forms of acute-phase SAA in circulation, [8]) by elisa indicated significant increase after HFHSC diet feeding compared to chow in both male (3.2-fold increase) and female (1.4-fold increase) WT mice (Fig 2E), consistent with increased systemic SAA in response to obesogenic diet. It was of interest to assess the specific induction of SAA3 after HFHSC feeding, since we [8] and others [42] previously reported that SAA3 is a major SAA isoform expressed in mouse adipose tissue during inflammation. Compared to plasma SAA1.1 and SAA2.1levels (Fig 2E; up to 200 μg/ml), SAA3 levels in the plasma were modest (Fig 2F; ~0.2–4.0 μg/ml), consistent with our earlier observation in mice injected with lipopolysaccharide [8]. There was no diet-induced increase in SAA3 in both male and female WT mice, as expected, SAA3 was undetectable in the plasma of all the TKO mice used in the experiment (Fig 2F). Consistently, SAA3 mRNA

abundance was significantly increased in the adipose tissues of male and female WT mice fed HFHSC diet compared to the chow-fed mice of the same sex, whereas SAA3 mRNA was at or below the level of detection in adipose tissues of all TKO mice (S1B Fig). The expression of the constitutively expressed SAA, SAA4 [43] was not significantly different in adipose tissues among the different groups of experimental mice (S1C Fig). SAA3 mRNA expression was also not significantly altered in the livers of WT mice after HFHSC diet feeding compared to chow (S1D Fig).

## Deficiency of SAA does not affect the development of diet-induced obesity

Compared with chow-fed mice, both male and female mice fed the HFHSC diet gained significantly more weight over the course of the study (Fig 3A and 3B), predominantly due to an increase in fat mass as measured by EchoMRI (Fig 3C). There was no significant difference in body weight or fat gain between the genotypes for either males or females after HFHSC diet feeding, indicating that the loss of SAA did not impact the development of obesity (Fig 3A–3C).

## Deficiency of SAA in HFHSC-fed mice has modest effects on glucose homeostasis

HFHSC diet caused a significant increase in fasting glucose levels in both WT and TKO male mice compared to chow diet (Fig 4A). HFHSC diet-fed male TKO mice had significantly increased fasting glucose levels than WT male mice fed the same diet (Fig 4A, left panel). In female mice, HFHSC diet feeding significantly increased fasting glucose levels only in TKO mice and not in WT mice compared to chow fed mice of the same genotype (Fig 4A, right panel). To investigate whether SAA deficiency impacts diet-induced glucose intolerance in the absence of alterations in adiposity, intraperitoneal glucose tolerance tests (GTT) were performed in male and female WT and TKO mice 12 weeks after initiation of diet feeding. For mice fed the normal rodent diet there was no impact of genotype on glucose excursions in either male or female mice (Fig 4B and 4C). As expected, mice fed HFHSC diet showed significantly impaired glucose tolerance compared to the corresponding chow fed mice, irrespective of their strain or sex (Fig 4B and 4C). Compared to WT mice, there was a modest but significantly increased impairment in glucose tolerance in both male (Fig 4B) and female (Fig 4C) TKO mice after HFHSC diet feeding.

To assess insulin sensitivity, the ability of insulin to decrease blood glucose concentration was determined by performing insulin tolerance tests (ITT) after 13 weeks of HFHSC feeding. There was no significant difference in insulin sensitivity with HFHSC diet in both male and female WT and TKO of the same sex (Fig 5A and 5B). There was also no difference in insulin tolerance between WT and TKO male (S2 Fig left panel) or female (S2 Fig right panel) mice fed chow diet (S2 Fig). Fasting insulin levels were significantly higher in HFHSC diet-fed WT and TKO male mice than that of chow-fed mice of the same strain and gender (Fig 5C, left panel). However, in female mice, HFHSC diet feeding for 14 weeks significantly increased fasting insulin levels only in TKO and not in WT mice compared to the chow-fed mice of the same strain (Fig 5C, right panel).

Glucose stimulated insulin secretion (GSIS) was assessed in WT and TKO mice fed HFHSC diet for 14 weeks by determining blood insulin levels before and 15 minutes after glucose injection in mice fasted for 16 h. There was no significant GSIS in male WT and TKO mice fed the HFHSC diet indicating diet-induced β-cell dysfunction (Fig 6, left panel). Both WT and TKO female mice showed significant increase in insulin secretion in response to glucose stimulation and there was no significant difference between the two strains (Fig 6, right panel).

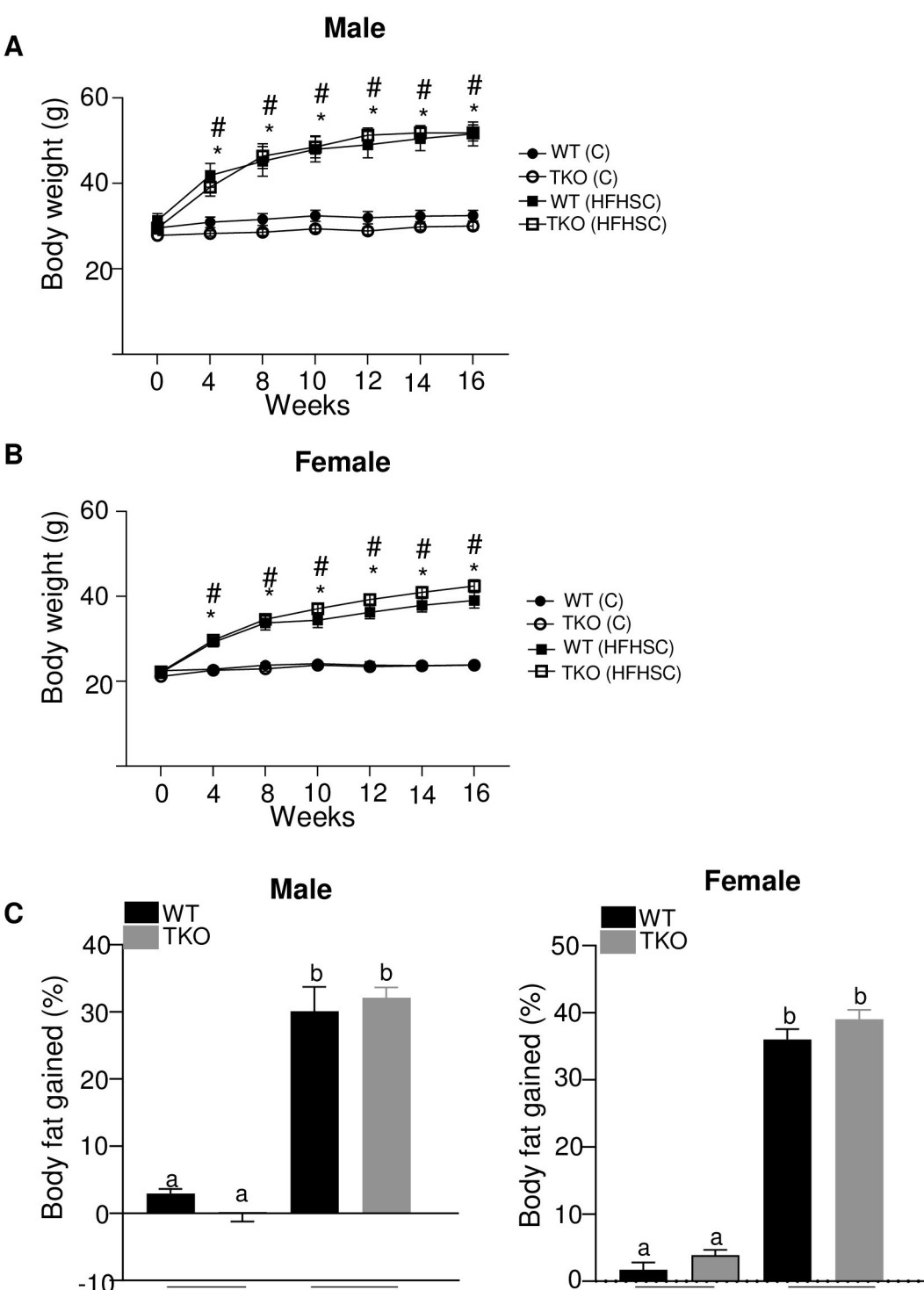

**Fig 3. Deficiency of SAA does not affect the development of diet-induced obesity.** A) body weight changes in male mice during the course of diet feeding B) body weight changes in female mice during the course of diet feeding C) percentage body fat gained in 15 weeks of chow and HFHSC diet feeding in male (left panel) and female mice (right panel). n = 5–15 mice/ group; Data are mean ±SEM; A and B: * and # = P ≤ 0.05 between WT (C) and WT (HFHSC) and between TKO (C) and TKO (HFHSC) respectively. In C, groups that are not significantly different (p ≥ 0.05) are indicated with the same letter.

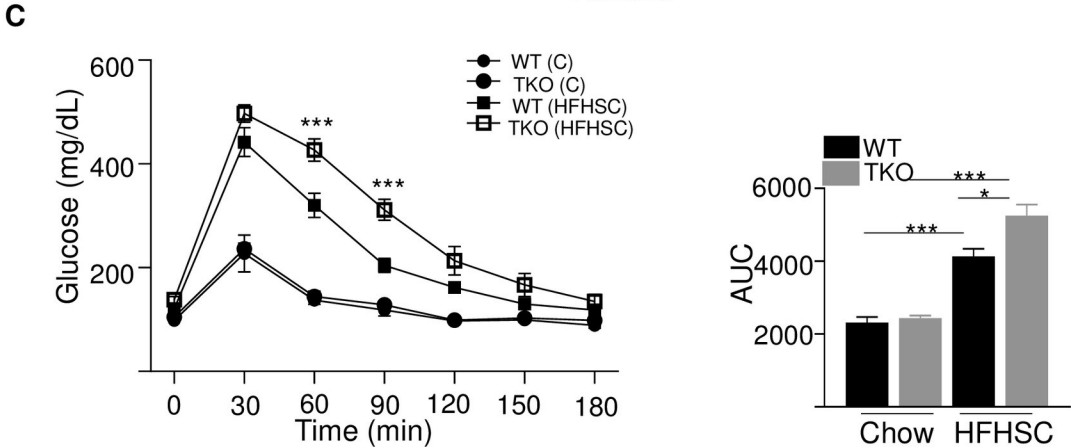

**Fig 4. Loss of SAA has a significant but modest effect on diet-induced glucose tolerance.** A) Basal glucose levels in male and female WT and TKO mice after 12 weeks of chow and HFHSC diet feeding (n = 5–15). Data are mean ±SEM; * = P ≤ 0.05; ** = P ≤ 0.01 and *** = p ≤ 0.001. B) intraperitoneal glucose tolerance test (IPGTT) was performed in male mice after 12 weeks of chow and HFHSC diet feeding (n = 10/group). The area under the curve (AUC) is shown in the right panel. Data are mean ±SEM; ** = P ≤ 0.01 between WT HFHSC and TKO HFHSC group of mice. * = P ≤ 0.05 and *** = P ≤ 0.001. C) IPGTT was performed in female mice after 12 weeks of chow and HFHSC diet feeding (n = 5-15/group). The area under the curve (AUC) is shown in the right panel. Data are mean ±SEM; *** = P ≤ 0.001 between WT HFHSC and TKO HFHSC group of mice. * = P ≤ 0.05 and ** = P ≤ 0.01.

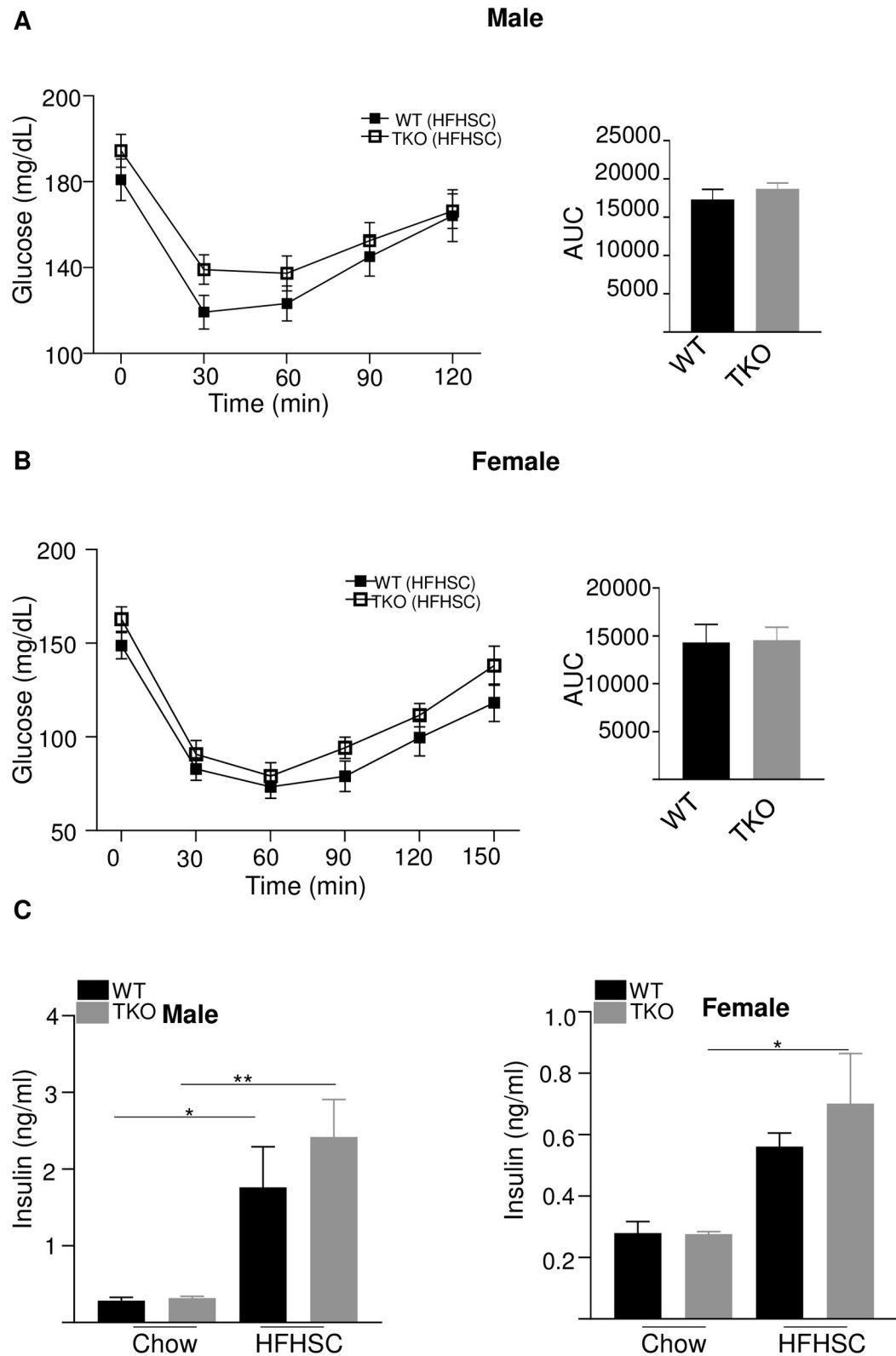

**Fig 5. Loss of SAA does not affect insulin tolerance or plasma fasting insulin levels.** A) insulin tolerance test (ITT) was performed in male WT and TKO mice after 13 weeks HFHSC diet (n = 10/group). AUC shown in the right panel B) insulin tolerance test (ITT) was performed in female WT and TKO mice after 13 weeks HFHSC diet (n = 10-15/group). AUC

shown in the right panel. C) basal plasma insulin levels in male (left panel) and female (right panel) mice fed either a chow or HFHSC diet for 14 weeks. Data are mean ±SEM; * = P ≤ 0.05 and ** = P ≤ 0.01.

## Deficiency of SAA does not impact diet-induced dyslipidemia or hepatic lipogenic gene expression

Earlier studies by Ather and Poynter [44] demonstrated that lack of SAA3 in mice promoted adult onset weight gain, hepatosteatosis and dyslipidemia in addition to significant weight gain with high-fat diet compared to WT litter controls. However, den Hartigh et al. [28] demonstrated that SAA3-deficient mice lost weight in response to an obesogenic diet and female SAA3-deficient mice but not male mice developed improved lipoprotein profiles and plasma lipid levels compared to WT litter controls. We investigated whether deficiency of SAA impacts plasma or liver lipid levels or hepatic lipogenic gene expression with or without HFHSC diet challenge. Fasting plasma triglyceride (TG) and plasma non-esterified fatty acids (NEFA) levels were similar for all groups of mice (S3A and S3B Fig). Plasma total cholesterol (TC) increased significantly with HFHSC diet in both WT and TKO mice, however there were no significant difference in TC between the two strains of mice (S3C Fig). Plasma HDL levels increased significantly with HFHSC diet in WT and TKO female but not male mice (S3D Fig). However, there were no significant differences in HDL between the two strains of male or female mice with either HFHSC or chow diet (S3D Fig). Liver TG increased significantly following HFHSC diet in both the strains of mice (S4A Fig), but there were no significant strain-dependent differences in liver TG with either chow or HFHSC diet (S4A Fig). Liver TC increased significantly with HFHSC diet only in male WT mice (S4B Fig); there were no significant changes in liver TC between any other groups of mice compared (S4B Fig). Hepatic lipogenic gene expression was analyzed in WT and TKO mice fed either chow or HFHSC diet for 16 weeks. There were no significant strain-specific

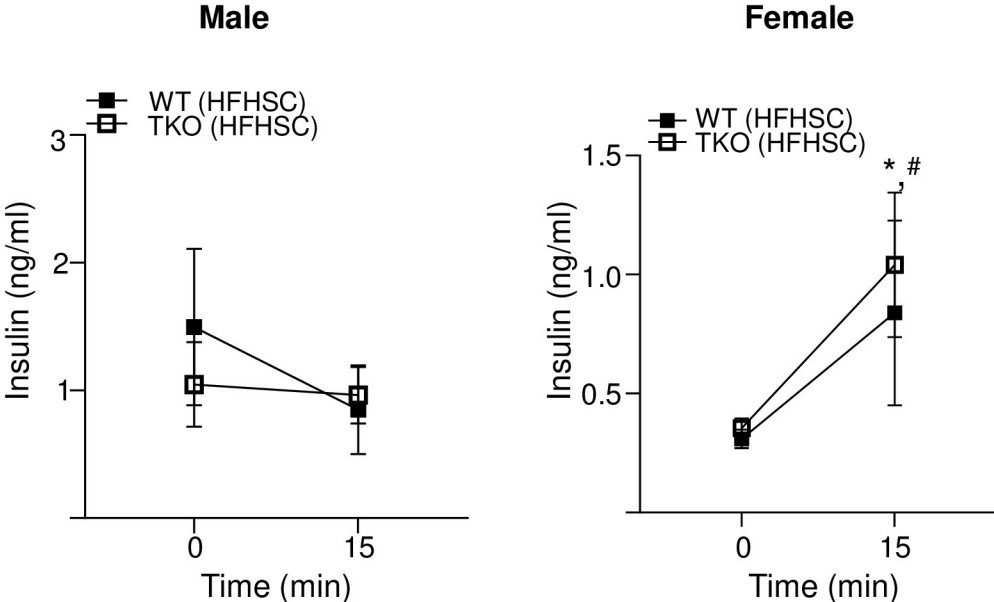

**Fig 6. Deficiency of SAA does not affect glucose stimulated insulin secretion.** Plasma insulin levels in male (left panel) and female (right panel) mice before and 15 minutes after intraperitoneal glucose injection, the mice were fed HFHSC diet for 14 weeks and fasted for 16 h prior to the study. Data are mean ±SEM;. * and # = P ≤ 0.05 between * WT chow and HFHSC diet and # between TKO chow and HFHSC diet.

differences in SREBP-1c, SCD-1, HMGCoAR or FAS mRNA abundance in livers of mice fed either chow or HFHSC diet (S5A–S5D Fig).

## Deletion of SAA does not affect adipose tissue inflammation in mice fed HFHSC diet

Adipose tissue inflammation and immune cell recruitment are hallmarks of obesity and play critical roles in the development of insulin resistance and glucose intolerance. Histological staining of gonadal fat sections from chow and HFHSC diet-fed WT and TKO mice showed no apparent differences in morphology between the two strains of mice. While there was a significant increase in adipocyte size with HFHSC diet feeding in both male and female mice. No significant strain-dependent differences were observed in female mice with either diet (Fig 7A and 7B).

Male and female mice displayed increased amounts of F4/80 mRNA in gonadal adipose tissue in response to HFHSC diet feeding (Fig 8A), indicating increased macrophage content in the obese mice. MCP-1, a chemokine implicated in the recruitment of inflammatory cells in obesity [45], was also increased in adipose tissue of mice fed HFHSC diet (Fig 8B). Moreover, F4/80 and MCP-1 mRNA abundance was similar in adipose tissue of WT and TKO mice fed the obesogenic diet indicating a lack of effect of SAA on obesity-associated adipose tissue inflammation (Fig 8A and 8B). Other indices of adipose tissue inflammation, including TNF-α (Fig 8C), PAI-1 (Fig 8D) were not significantly altered by SAA deficiency in either male or female mice after obesogenic diet. IL-6 or IL-1β mRNA levels did not change significantly irrespective of diet, gender or strain (S6A and S6B Fig). Plasma endotoxin levels were comparable among the different groups of mice (S6C Fig).

## Discussion

In this study, using male and female SAA-deficient mouse model (deficient in all three inducible SAA isoforms), we investigated if SAA has any role in the development of diet-induced obesity, adipose tissue inflammation, and impairment in glucose or lipid metabolism. The major finding from the study is that while SAA levels in adipose tissue, liver and in circulation are markedly increased with obesity, deficiency of SAA does not impact the development of obesity or obesogenic diet-induced adipose tissue inflammation. Contrary to the thought that SAA may promote obesity related impairment in glucose homeostasis, the present study shows a modest worsening of glucose tolerance in SAA-deficient mice (TKO) compared to the WT mice on HFHSC diet. However, there was no significant effects on insulin tolerance or GSIS. SAA does not contribute to diet-induced dyslipidemia or altered liver lipid homeostasis. Thus, taken together, our study confirms that in mice, the expression of inducible SAA isoforms is significantly increased in liver and adipose tissues after obesogenic diet feeding. However, this induction appears to be a consequence, not a cause, of diet-induced obesity and adipose tissue inflammation.

A number of studies in the past have shown altered SAA levels in circulation and in adipose tissues with changes in diet and body weight, raising the possibility that SAA plays a role in the development of obesity and/or obesity-related complications [15–18]. Studies by Sjoholm and colleagues suggest a sexual dimorphic role of SAA in obesity and obesity-associated inflammation [46]. *In vivo* studies have provided contradictory reports on the roles of SAA in obesity and associated metabolic complications. For example, some authors have reported that SAA3 suppresses adult onset and diet-induced weight gain. However, contrary to this conclusion, other authors have reported that SAA3 exacerbates weight gain induced by an obesogenic diet in a sexually dimorphic manner [28]. Vercalsteren et al. reported that SAA3 gene silencing impaired adipogenesis by studies *in vitro* in murine preadipocyte cell lines (3T3-F442A), when SAA3-silenced preadipocytes were implanted into BALB/c Nude mice, the mice developed smaller fat pads than their

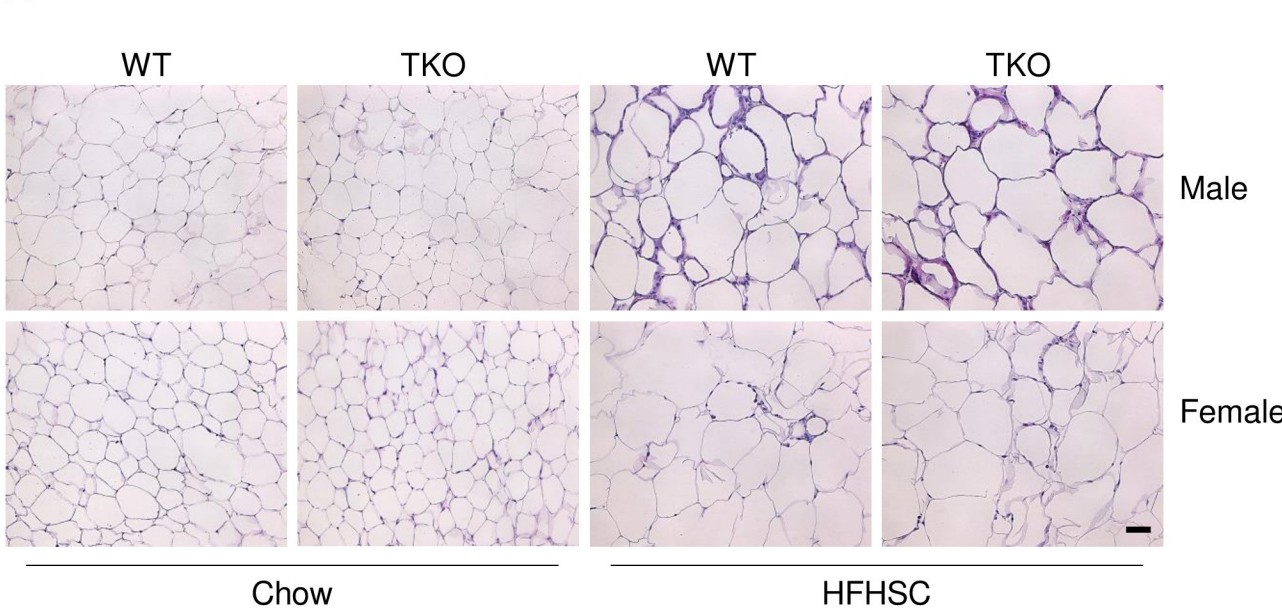

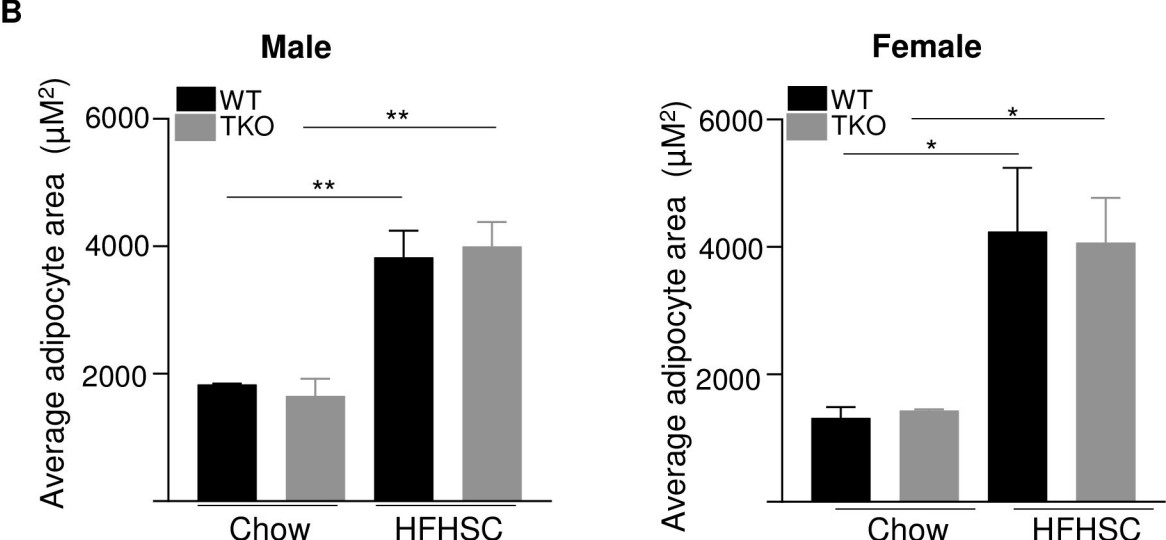

**Fig 7. Deficiency of SAA does not affect diet-induced changes in adipose tissue structure.** A) Representative images of HE staining in gonadal white adipose tissues in male and female WT and TKO mice fed either chow or HFHSC diet for 16 weeks. Scale bar 50 μm. B) adipocyte area was determined for 3 randomly chosen sections for 3 mice/strain; male (left) and female (right) mice; data are mean ±SEM;. * and ** = P ≤ 0.05 and ≤ 0.01 respectively.

control counterparts [47]. Silencing SAA1.1 and SAA2.1 expression in Swiss Webster mice by antisense oligonucleotides was reported to reduce adipose tissue expansion significantly in these mice along with suppressed adipose tissue inflammation and improved glucose and insulin tolerance in mice fed high fat diet [30]. Differences in body weight or body fat gain between WT and TKO mice fed either chow or HFHSC diet were not observed in the current study (Fig 3A–3C). Phenotypic differences observed between this study and the earlier studies [28, 44, 47] could be either due the differences in mouse strains used, mice with deficiency of all three SAA isoforms vs

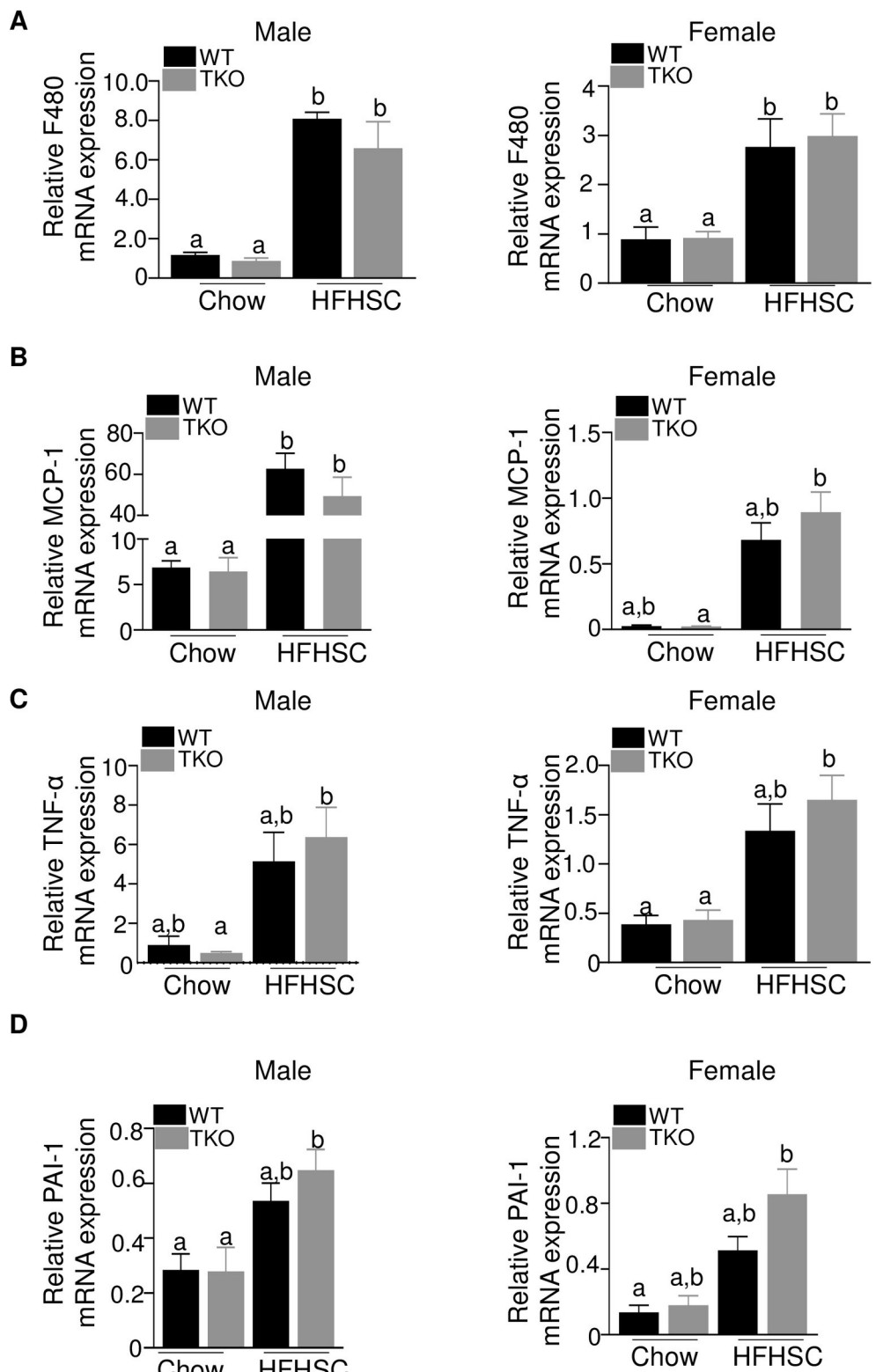

**Fig 8. Deficiency of SAA does not affect diet-induced changes in adipose tissue inflammatory status.** A-D) Expression of F480, MCP-1, TNFα and PAI-1genes respectively in the gonadal adipose tissues of male and female WT and TKO mice fed either chow or HFHSC diet for 16 weeks (n = 4-10/group). Data are mean ±SEM; data that are not significantly different (P>0.05) are indicated with the same letter.

mice with SAA3 or SAA1.1/2.1 deficiency or the diet used. SAA3 is considered a pseudogene in humans [7] hence the results from the studies on SAA3 have no relevance to humans. However, the presence of SAA3 cannot be ignored in animal studies due its possible biological effects that may or may not overlap with SAA1.1 and SAA2.1 [25]. To our knowledge this is the only study where mice deficient in all the three inducible isoforms of SAA were used to investigate the role of SAA in obesity and associated metabolic complications.

*In vitro* studies have also implicated SAA in obesity and obesity-associated metabolic perturbations. Incubating 3T3-L1 preadipocytes with recombinant hSAA1 was shown to result in enhanced proliferation, decreased differentiation and altered insulin sensitivity [48]. Results from several *in vitro* studies could not be reproduced when tested *in vivo*. For example, a number of *in vitro* studies show SAA to have pro-inflammatory properties (reviewed in [7, 49]), however, such activities were not observed with manipulation of SAA expression *in vivo* [21, 50]. Several *in vitro* studies have used commercially available recombinant SAA which has two amino acid substitutions (at positions 61 and 71) compared to native and thus may show activities different from either mouse or human SAA [51, 52]. Discrepancies in properties between recombinant SAA and endogenous SAA purified from acute-phase plasma have been shown *in vitro* [25]. Possible lipopolysaccharide contamination of E. coli-derived recombinant SAA could contribute to some of the proinflammatory activities exhibited by this protein [53]. Hence results from all papers published using recombinant SAA for *in vitro* studies should be reinterpreted with caution. Furthermore, it is now recognized that activities observed for lipid-free SAA *in vitro* are lost when SAA is lipid-associated, the form SAA is believed to exist *in vivo* [21, 26, 54].

The existence of multiple isoforms also poses a challenge to SAA studies. Potential functional differences between the three murine acute phase SAA isoforms have not be comprehensively examined. Though mouse SAA1.1 and 2.1 share 91% protein sequence identity [25], differences in properties of the two isoforms have been observed in its amyloidogenic potential [55]. Mouse SAA3 shares ~67% sequence identity to mouse SAA1.1 and SAA2.1 [25], yet differs from the other two with regard to its expression profile and HDL association [8]. Further, there exists a less studied minor acute-phase [56] or a constitutive [11, 57] isoform of SAA, SAA4, which contributes to more than 90% of total circulating SAAs in the absence of inflammation [58]. Mouse models deficient in all four isoforms of SAA including SAA4 have been developed recently and studied in the context of retinol-binding and adaptive immunity [59–61]. The contributions of SAA4 if any, to obesity and glucose homeostasis need to be investigated in future. Our data indicate that deletion of all inducible SAA isoforms, including SAA3, does not impact weight gain, adipose tissue inflammation, or major metabolic consequences that occur in mice with obesogenic diet feeding. Whether deletion of individual isoforms of SAA produce different effects cannot be ruled out at this time and is a limitation of our study.

Based on human population studies, elevated levels of circulating SAA have been recognized as a risk factor for certain chronic inflammatory diseases for many decades, including atherosclerotic cardiovascular disease [49] and cancer [62]. The development of mice with targeted deletion of the inducible SAAs allowed us to identify a causative role for SAA in both atherosclerosis [32, 36] and metastatic pancreatic cancer. Here we report that the loss of all three inducible SAAs has no effect on the development of diet-induced obesity or adipose tissue inflammation, with a modest but significant protective effect on glucose intolerance. Thus, increased SAA in the setting of obesity appears to be a consequence, not a cause, of adipose tissue inflammation.

## Supporting information

**S1 Fig.** A) Genotyping to identify SAA TKO mice are done by a multi-step process. The first step is genotyping for SAA1/2 KO utilizing a 3 primer PCR reaction as described earlier (de

Beer et al., J Lipid Res. 2010. 51:3117–3125). The genotype to determine SAA 3 deficiency is performed by two sets of PCRs, using forward primer for the PCRs designed to detect WT sequence or SAA3 KO (CRISPR-Cas9) sequences respectively. The primers for the two PCRs are shown on the right panel. The generation of TKO mice from SAA1.1/SAA2.1-deficient mice is described earlier [35]. B) Expression of SAA3 mRNA in the adipose tissues of male (left panel) and female (right panel) WT and TKO mice fed either chow or HFHSC diet for 16 weeks. C) SAA4 mRNA expression in the adipose tissues of male (left panel) and female (right panel) WT and TKO mice fed either chow or HFHSC diet for 16 weeks. D) Expression of SAA3 mRNA in the livers of male (left panel) and female (right panel) WT and TKO mice fed either chow or HFHSC diet for 16 weeks. Data are mean ±SEM; $^{**}$ = P $\leq$ 0.01 and $^{***}$ = P $\leq$ 0.001.
(PPTX)

**S2 Fig. Deficiency of SAA does not affect basal insulin tolerance in mice.** Insulin tolerance test (ITT) was performed in male (left panel) and female (right panel) WT and TKO mice on chow diet (n = 4–5 mice/group).
(PPTX)

**S3 Fig. Deficiency of SAA does not significantly impact diet-induced dyslipidemia.** Plasma triglycerides A), non-esterified fatty acids B) and total cholesterol C) levels in male (left panel) and female (right panel) WT and TKO mice (n = 4–15) fed either chow or HFHSC diet for 16 weeks. D) Plasma HDL levels in male (left panel) and female (right panel) WT and TKO mice (n = 4–15) fed either chow or HFHSC diet for 16 weeks. Data are mean ±SEM; data that are not significantly different (P>0.05) are indicated with the same letter.
(PPTX)

**S4 Fig. Deficiency of SAA does not significantly impact diet-induced changes in hepatic triglycerides or cholesterol levels.** Liver triglycerides A) and total cholesterol B) levels in male (left panel) and female (right panel) WT and TKO mice (n = 5–15) fed either chow or HFHSC diet for 16 weeks. Data are mean ±SEM; data that are not significantly different (P>0.05) are indicated with the same letter.
(PPTX)

**S5 Fig. Deficiency of SAA does not significantly impact diet-induced changes in hepatic lipogenic gene expression.** Expression of SREBP1-c, SCD-1, HMGCoAR and FAS genes (A-D) respectively in the livers of male (left panel) and female (right panel) WT and TKO mice fed either chow or HFHSC diet for 16 weeks (n = 4-10/group). Data are mean ±SEM; data that are not significantly different (P>0.05) are indicated with the same letter.
(PPTX)

**S6 Fig. Deficiency of SAA does not significantly impact diet-induced changes in adipose tissue IL-6 and IL-1β gene expression levels.** Expression of IL-6 A) and IL-1β B) mRNA in the adipose tissues of male (left panel) and female (right panel) WT and TKO mice fed either chow or HFHSC diet for 16 weeks (n = 4-10/group). C) Plasma endotoxin levels at the study termination in male (left panel) and female (right panel) WT and TKO mice fed either chow or HFHSC diet for 16 weeks (n = 4-10/group). Data are mean ±SEM.
(PPTX)

**S1 Raw images.**
(PDF)

**S2 Raw images.**
(PDF)

**S1 Data.**
(XLSX)

## Author Contributions

**Conceptualization:** Nancy R. Webb, Lisa R. Tannock, Preetha Shridas.

**Data curation:** Ailing Ji, Preetha Shridas.

**Formal analysis:** Ailing Ji, Preetha Shridas.

**Funding acquisition:** Frederick C. De Beer, Nancy R. Webb, Lisa R. Tannock, Preetha Shridas.

**Investigation:** Ailing Ji, Andrea C. Trumbauer, Victoria P. Noffsinger, Hayce Jeon, Avery C. Patrick, Preetha Shridas.

**Methodology:** Ailing Ji, Nancy R. Webb, Lisa R. Tannock, Preetha Shridas.

**Project administration:** Preetha Shridas.

**Resources:** Frederick C. De Beer, Nancy R. Webb, Lisa R. Tannock.

**Supervision:** Preetha Shridas.

**Validation:** Ailing Ji, Preetha Shridas.

**Visualization:** Ailing Ji, Preetha Shridas.

**Writing – original draft:** Preetha Shridas.

**Writing – review & editing:** Ailing Ji, Nancy R. Webb, Lisa R. Tannock.

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
