## [Decision Letter · Decision Letter 0]

15 Nov 2021

PONE-D-21-32072Serum Amyloid A is not obligatory for high-fat, high-sucrose, cholesterol-fed diet-induced obesity and its metabolic and inflammatory complicationsPLOS ONE

Dear Dr. Shridas,

Thank you for submitting your manuscript to PLOS ONE. After careful consideration, we feel that it has merit but does not fully meet PLOS ONE’s publication criteria as it currently stands. Therefore, we invite you to submit a revised version of the manuscript that addresses the points raised during the review process.

We look forward to receiving your revised manuscript.

Kind regards,

Michael Bader

Academic Editor

PLOS ONE

Journal Requirements:

- https://onlinelibrary.wiley.com/doi/10.1002/oby.20126

- https://link.springer.com/article/10.1007%2Fs11883-020-00901-4

The text that needs to be addressed involves the introduction

In your revision ensure you cite all your sources (including your own works), and quote or rephrase any duplicated text outside the methods section. Further consideration is dependent on these concerns being addressed.

Reviewers' comments:

Reviewer's Responses to Questions

**Comments to the Author**

1. Is the manuscript technically sound, and do the data support the conclusions?

Reviewer #1: Yes

Reviewer #2: Yes

Reviewer #3: Yes

2. Has the statistical analysis been performed appropriately and rigorously? 

Reviewer #1: Yes

Reviewer #2: Yes

Reviewer #3: Yes

3. Have the authors made all data underlying the findings in their manuscript fully available?

Reviewer #1: Yes

Reviewer #2: Yes

Reviewer #3: Yes

4. Is the manuscript presented in an intelligible fashion and written in standard English?

Reviewer #1: Yes

Reviewer #2: Yes

Reviewer #3: Yes

5. Review Comments to the Author

Reviewer #1: The manuscript by Ji et al. explores the contribution of serum amyloid A (SAA) to obesity generation and pathophysiology. They document the diet-induced obesity feeding regimen (high fat, sucrose, and cholesterol) induces SAA expression, but that mice deficient in SAA do not display any substantial or biologically meaningful differences as a consequence of feeding of this diet compared to wildtype mice (an exception perhaps being glucose levels). These results substantiate a lack of involvement of SAAs in diet-induced obesity and associated pathophysiological manifestations. The manuscript is straightforward and the results appropriately presented. However, there are areas of the work that require additional attention in a revised version of the manuscript.

First, critically absent are any data documenting the purported genotype, gene expression absence, and protein abundance decreases in the SAA triple knockout mice. Similar mice have been previously published (PMID 31484771, which are SAA1,2,3,4-deficient), yet the mice used in this manuscript appear to have been generated independently. Consequently, the authors need to present compelling evidence that their mice are what they are claimed to be, and that is mice absent of SAA1, SAA2, and SAA3 proteins. Having never encountered ELISAs that recognize all three SAAs (SAA1/2 are oftentimes both recognized by an SAA ELISA, whereas SAA3 is not unless it is SAA3-specific), several assays will need to be conducted. Furthermore, the aforementioned citation should be included and the new mouse compared to that one.

Second, the authors cited a paper related to the spontaneous development of obesity in SAA3-/- mice fed normal chow (PMID 29390039), but only in the contest of one figure in which it was reported that feeding a HFD for 1 week to the SAA3-/- mice increased weight gain compared to wildtype mice. That was one figure in the paper that otherwise indicated spontaneous obesity in the SAA3-/- mice that was conducted using a different HFD than that used in this manuscript. Additional clarification of these differences should be conveyed by the authors.

Finally, the authors cite papers in which 3T3L1 cells were treated with recombinant SAA, which is problematic due to the fact that recombinant forms of SAA that are generated from E. coli contain contaminants (including TLR2 stimulating lipopeptides). The authors need to mention these findings (PMID) and the citations in that paper that have long suggested a lack of inflammatory activity of SAA proteins. All papers published using recombinant SAA proteins should be reinterpreted with caution, and acknowledgement of the contamination issue needs to be understood by researchers and conveyed by authors in their publications.

Reviewer #2: I think this study adds another piece on the path to elucidating the biological effects of SAA. Some questions have been repeatedly asked: Can the results observed with recombinant SAA be extrapolating to in vivo? Are there free SAA stocks? How much? In serum? in tissues? How do the different forms of SAA work? Is there a regulation between them? How do differences between the biochemistry and metabolism of lipoproteins between species affect the SAA role? Is it possible to extrapolate results from experimental animals to humans? How to establish causal relationships between SAA increase and decrease with some of the effects pointed out for it?

In this complex scenario, the study by Ailing et al. has credit and deserves to be published in PlosOne. The results with the triple deficient in SAA are fair and the experimental design well delineated.

The comparison between males and females is also all interesting.

My comments are:

1- Do the authors have data on food consumption, serum HDL, and endotoxemia comparing TKO animals with WT? If there is any possibility to include these data it will be interesting.

2- It would also have been fantastic to have the data for the single and double deletions, in addition to the triple deletion..... same for super expression.. But I understand perfectly well that this is a mid-term mission.

3- Anyway, I consider that some of these points might be discussed by the authors. Beyond the limitations of previous studies, adding their study limits will provide a clearer picture and put in context the complexity of SAA biology.

Reviewer #3: General comments:

The first sentence of the abstract doesn’t give an appropriate sense of what this manuscript is about, i.e., whether SAA plays a role in the development of obesity and adipose tissue inflammation. It and the second sentence of the abstract should be revised accordingly. The manuscript purports to challenge one proposed role of SAA in obesity, i.e., its role in causing obesity, which is not widely universally accepted based on the limited number of studies. However, the findings in this study show that SAA deletion doesn’t affect the development of obesity and hence doesn’t challenge this proposed role of SAA, which is not well-established. Nor do the finding challenge the findings that SAA levels, a measure of the modest obesity-driven inflammation, is increased as a consequence of obesity. This finding is firmly established in many studies and is confirmed in this study (Fig 2). The portion of the abstract concerning challenging the proposed role of SAA in obesity is somewhat misleading and should be revised accordingly.

The authors quote several studies that supposedly support the notion that SAA might be involved in the development of obesity and its consequences., yet most of these indicate that they affect the production of cytokines, ROS, nitric oxide or inflammatory cell recruitment, rather than the development of obesity per se. However, the quote an article that shows that suppression of SAA by ASOs caused a reduction in adipose tissue expansion and another in which SAA3 deletion blunted diet-induced weight gain in in female, suggesting that SAA might play a role in the development of obesity in addition to inflammation. Nonetheless, the notion that SAA plays a role in the pathogenesis of obesity is not well established.

Their study purports to determine whether “SAA plays a functional role in the development of adipose tissue inflammation, insulin resistance and other metabolic complications, or is a mere marker of inflamed adipose tissue”. However, these two are not mutually exclusive and the way the manuscript is written doesn’t clearly distinguish between the possibilities that SAA is responsible for the development of obesity and obesity-associated inflammation and that SAA and obesity-associated inflammation is a result of obesity. Their findings show that deletion of SAA1.1, 2.1 and 3 (i.e., not all isotypes as stated elsewhere since SAA4 was apparently intact), does not affect the development of diet induced obesity, hepatic lipids or adipose tissue inflammation, although there was a modest effect of glucose metabolism. In other words, it was a largely negative study, and as such doesn’t provide evidence that SAA is a cause of obesity and obesity-associated inflammation. Nor does it really challenge much of the current thinking of the role of SAA resulting from the development of obesity. Their conclusions need to state this more succinctly.

Specific points:

There is currently no model validation in this paper. The authors must show clear evidence of SAA1,2,3 deficiency across multiple tissues (liver and EWAT, at a minimum). This should be added to Figure 2. The original reference to the TKO mice (reference 30) does not show that SAA3 is knocked out from WAT, or that any SAA subtypes are knocked out from liver. Also, the other reference they listed for the TKO mouse model (28) is incorrect. Measurement of plasma SAA levels also should be provided for the TKO mice with an without the HFHSC diet as further model validation .

The age at which the mice were started on the high fat diet should be stated.

The authors should quantify adipocyte size. Based on the images presented in Fig. 7, it looks as if the HFHSC-fed TKO mice had larger adipocytes.

The authors only included 5 mice per group. This seems very small - was a power calculation performed to determine that 5 mice/group would be sufficient?

The discussion section is too short and should be expanded to include the issues brought up in the general section of this review.

6. PLOS authors have the option to publish the peer review history of their article (what does this mean?). If published, this will include your full peer review and any attached files.

Reviewer #1: No

Reviewer #2: **Yes: **Ana Campa

Reviewer #3: No

---

## [Author Response · Author response to Decision Letter 0]

28 Jan 2022

Response to Reviewer’s comment:

We thank the reviewers for their positive comments recognizing that the manuscript entitled “Serum Amyloid A is not obligatory for high-fat, high-sucrose, cholesterol-fed diet-induced obesity and its metabolic and inflammatory complications” (PONE-D-21-32072) has merit to be published in PLOS ONE. As requested, we provide a point-to-point response to all reviewer’s comments below. Changes in the resubmitted manuscript are indicated as yellow highlighted text with red color font (please see ’Revised manuscript with Track Changes’).

New and revised figures are described in our point-to-point response, below. 

In response to the suggestion, we have ensured that the manuscript meets PLOS ONE’s style requirements. 

- https://onlinelibrary.wiley.com/doi/10.1002/oby.20126

- https://link.springer.com/article/10.1007%2Fs11883-020-00901-4

In your revision ensure you cite all your sources (including your own works), and quote or rephrase any duplicated text outside the methods section. Further consideration is dependent on these concerns being addressed.

We thank the editor for pointing out our oversight. In response to the comment we have now carefully rephrased duplicated text and also included appropriate citations (introduction, page number 3, lines 1-3). 

We thank the editor for pointing out the mistake. We have corrected the financial information and Financial Disclosure sections in the revised manuscript. Under acknowledgement, we have acknowledged the use of facilities and resources provided by the Centers of Biomedical Research Excellence (COBRE) at the University of Kentucky core which was supported by an Institutional Development Award (IDeA) from the National Institute of General Medical Sciences of the National Institutes of Health under grant number P30 GM127211. However, the grant did not provide any funding for the study. 

4. Data availability 

The data is provided in the “Raw data” file. 

Experimental results from the insulin tolerance tests (ITT) performed in WT and TKO mice fed chow diet are now included as supplemental Figure (S2 Figure). The phrase “data not shown” has been removed from the ‘Results’ section (page #11, lines 18-19). 

Response to reviewer’s comment

Reviewer 1:

1.First, critically absent are any data documenting the purported genotype, gene expression absence, and protein abundance decreases in the SAA triple knockout mice. Similar mice have been previously published (PMID 31484771, which are SAA1,2,3,4-deficient), yet the mice used in this manuscript appear to have been generated independently. Consequently, the authors need to present compelling evidence that their mice are what they are claimed to be, and that is mice absent of SAA1, SAA2, and SAA3 proteins. Having never encountered ELISAs that recognize all three SAAs (SAA1/2 are oftentimes both recognized by an SAA ELISA, whereas SAA3 is not unless it is SAA3-specific), several assays will need to be conducted. 

Although the SAA triple knockout mice have been studied in previous publications by our lab and our collaborators’ lab (Thompson et al., J Lipid Res. 2015;56(2):286-93; Lee et al., Cell. 2020;180(1):79-91 e16), we recognize that rigorous studies confirming the genotype of these mice has not yet been published. Therefore, in response to the reviewer’s comment, we include the following several data to show the lack of detectable SAA 1,2 and 3 mRNA and protein expression in the TKO mice: 

1. We have included results from Western blot analysis, Figure 2B, to demonstrate lack of SAA protein expression in adipose tissue extracts of TKO mice. We mixed two different antibodies in the western blot which were validated (Tannock et al., 2018; J Lipid Res, 59:339) to specifically recognize SAA1,2 and SAA3 respectively.

2. We have included new figure 2C to demonstrate lack of SAA mRNA expression in livers of male and female TKO mice fed either chow or HFHSC diet. 

3. We have included new figure 2D, which shows results from western blot analysis of plasma from experimental mice and demonstrates lack of detectable SAA protein in the TKO mice. The antibody used recognizes all three isoforms of SAA, SAA1,2 and 3. We mixed two different antibodies which were validated (Tannock et al., 2018; J Lipid Res, 59:339) to specifically recognize SAA1,2 and SAA3 respectively.

4. We have included new supplemental figures that assess SAA3 mRNA abundance in adipose tissue (S1B) and livers (S1D) of male and female WT and TKO mice after chow and HFHSC feeding. As demonstrated in S1C, we also show that SAA4 mRNA expression in the adipose tissues is not significantly altered in the TKO mice compared to WT. 

5. We have also included the details of how genotyping is done to validate the strains of the mice used in the revised manuscript. The genotyping is done in two steps, the first step to validate the absence of SAA1 and 2 (de Beer et al., J Lipid Res. 2010. 51:3117-3125) and the second to validate the absence of SAA3 (S1A supplement).

Furthermore, the aforementioned citation (PMID 31484771, which are SAA1,2,3,4-deficient) should be included and the new mouse compared to that one.

In response to the Reviewer’s comment, we have included the citation and compared our mouse model to that one (Discussion, page 16, lines 17-18).

2. Second, the authors cited a paper related to the spontaneous development of obesity in SAA3-/- mice fed normal chow (PMID 29390039), but only in the contest of one figure in which it was reported that feeding a HFD for 1 week to the SAA3-/- mice increased weight gain compared to wildtype mice. That was one figure in the paper that otherwise indicated spontaneous obesity in the SAA3-/- mice that was conducted using a different HFD than that used in this manuscript. Additional clarification of these differences should be conveyed by the authors.

We thank the reviewer for pointing out our oversight in reporting the results from a previously published paper ((PMID 29390039). In response to the reviewer’s comment, we have now included the results of the previously published study in Results, under the heading “Deficiency of SAA does not impact diet-induced dyslipidemia or hepatic lipogenic gene expression”, (lines 1-3) and also we have discussed the discrepancies observed between the different reports in the “Discussion”, page 15, lines 5-21 (highlighted text). 

3. Finally, the authors cite papers in which 3T3L1 cells were treated with recombinant SAA, which is problematic due to the fact that recombinant forms of SAA that are generated from E. coli contain contaminants (including TLR2 stimulating lipopeptides). The authors need to mention these findings (PMID) and the citations in that paper that have long suggested a lack of inflammatory activity of SAA proteins. All papers published using recombinant SAA proteins should be reinterpreted with caution, and acknowledgement of the contamination issue needs to be understood by researchers and conveyed by authors in their publications.

We thank the reviewer for pointing out the need to describe the discrepancies in properties observed in studies using E. coli-derived recombinant SAA compared to endogenous/purified SAA preparation. We have discussed the issue in our ‘Discussion’, page 15, second paragraph lines 3-12 of the revised manuscript (high-lighted text). 

Reviewer 2

1. Do the authors have data on food consumption, serum HDL, and endotoxemia comparing TKO animals with WT? If there is any possibility to include these data it will be interesting.

We did not measure food intake in the current study. However, there were no overt differences in food consumption between the two strains of male or female mice during the experiment. In a previous unpublished study, WT and TKO male and female mice were fed a high fat diet (60% fat diet) and placed in acclimation chambers for 1 week and recorded in indirect colorimetry system (TSE-Systems Inc., Chesterfield, MO) for 1 week. No differences in food consumption between WT and TKO mice were observed (shown above is the data from male mice). 

We have now included new data showing plasma HDL levels in male and female WT and TKO mice after chow and HFHSC diet feeding (figure S3D). A summary of the results is now included in the “Results” section of the revised manuscript under the heading “Deficiency of SAA does not impact diet-induced dyslipidemia or hepatic lipogenic gene expression”,page 12, lines 7-9 (highlighted text). 

We have now added new data showing plasma endotoxin levels in the experimental mice (new figure, S6C) and included the results in “Results” section, under the heading “Deletion of SAA does not affect adipose tissue inflammation in mice fed HFHSC diet”, page 14, line 7 (highlighted text).

1- It would also have been fantastic to have the data for the single and double deletions, in addition to the triple deletion..... same for super expression.. But I understand perfectly well that this is a mid-term mission.

Thank you for understanding our limitations in performing experiments with single and double deletion mice. 

3- Anyway, I consider that some of these points might be discussed by the authors. Beyond the limitations of previous studies, adding their study limits will provide a clearer picture and put in context the complexity of SAA biology.

In response to reviewer’s suggestion, we have now included points regarding the possible role of individual SAAs in our “Discussion”, page 16, 2nd paragraph, lines 1-13 (highlighted text).

Reviewer 3:

1. The first sentence of the abstract doesn’t give an appropriate sense of what this manuscript is about, i.e., whether SAA plays a role in the development of obesity and adipose tissue inflammation. It and the second sentence of the abstract should be revised accordingly. The manuscript purports to challenge one proposed role of SAA in obesity, i.e., its role in causing obesity, which is not widely universally accepted based on the limited number of studies. However, the findings in this study show that SAA deletion doesn’t affect the development of obesity and hence doesn’t challenge this proposed role of SAA, which is not well-established. Nor do the finding challenge the findings that SAA levels, a measure of the modest obesity-driven inflammation, is increased as a consequence of obesity. This finding is firmly established in many studies and is confirmed in this study (Fig 2). The portion of the abstract concerning challenging the proposed role of SAA in obesity is somewhat misleading and should be revised accordingly.

We agree with the reviewer that the abstract does not appropriately reflect the contribution of our study to the field. As suggested we have now modified the abstract in the revised manuscript (highlighted text).

2. The authors quote several studies that supposedly support the notion that SAA might be involved in the development of obesity and its consequences., yet most of these indicate that they affect the production of cytokines, ROS, nitric oxide or inflammatory cell recruitment, rather than the development of obesity per se. However, the quote an article that shows that suppression of SAA by ASOs caused a reduction in adipose tissue expansion and another in which SAA3 deletion blunted diet-induced weight gain in in female, suggesting that SAA might play a role in the development of obesity in addition to inflammation. Nonetheless, the notion that SAA plays a role in the pathogenesis of obesity is not well established.

We thank the reviewer for the suggestions, we have now modified the abstract and conclusions to indicate that SAA is the consequence of obesity-associated inflammation and does not contribute to the development of it (abstract first and last three lines; discussion last paragraph highlighted text). 

Their study purports to determine whether “SAA plays a functional role in the development of adipose tissue inflammation, insulin resistance and other metabolic complications, or is a mere marker of inflamed adipose tissue”. However, these two are not mutually exclusive and the way the manuscript is written doesn’t clearly distinguish between the possibilities that SAA is responsible for the development of obesity and obesity-associated inflammation and that SAA and obesity-associated inflammation is a result of obesity. Their findings show that deletion of SAA1.1, 2.1 and 3 (i.e., not all isotypes as stated elsewhere since SAA4 was apparently intact), does not affect the development of diet induced obesity, hepatic lipids or adipose tissue inflammation, although there was a modest effect of glucose metabolism. In other words, it was a largely negative study, and as such doesn’t provide evidence that SAA is a cause of obesity and obesity-associated inflammation. Nor does it really challenge much of the current thinking of the role of SAA resulting from the development of obesity. Their conclusions need to state this more succinctly.

Specific points:

There is currently no model validation in this paper. The authors must show clear evidence of SAA1,2,3 deficiency across multiple tissues (liver and EWAT, at a minimum). This should be added to Figure 2. The original reference to the TKO mice (reference 30) does not show that SAA3 is knocked out from WAT, or that any SAA subtypes are knocked out from liver. Also, the other reference they listed for the TKO mouse model (28) is incorrect. Measurement of plasma SAA levels also should be provided for the TKO mice with an without the HFHSC diet as further model validation.

We thank the reviewer for pointing out the requirement for model validation in the paper. Accordingly, we have added several new data to demonstrate the absence of SAA expression in the circulation, adipose tissues as well as in the liver. 

The authors must show clear evidence of SAA1,2,3 deficiency across multiple tissues (liver and EWAT, at a minimum).

In response to this comment, we have included the following new data, the figures showing the absence of SAA (all the three inducible isoforms) expression in gonadal adipose tissues and liver are now added to figure 2 of the revised manuscript. In addition, we have also included data showing the absence of SAA3 specifically in adipose tissues and livers of the deficient mice:

1. We have included results from Western blot analysis, Figure 2B, to demonstrate lack of SAA protein expression in subcutaneous adipose tissue extracts of TKO mice. We mixed two different antibodies in the western blot which were validated (Tannock et al., 2018; J Lipid Res, 59:339) to specifically recognize SAA1,2 and SAA3 respectively.

2. We have included new figure 2C to demonstrate lack of SAA mRNA expression in livers of male and female TKO mice fed either chow or HFHSC diet. 

3. We have included new supplemental figures that assess SAA3 mRNA abundance in adipose tissue (S1B) and livers (S1D) of male and female WT and TKO mice after chow and HFHSC feeding. As demonstrated in S1C, we also show that SAA4 mRNA expression in adipose tissues is not significantly altered in the TKO mice compared to WT. 

The original reference to the TKO mice (reference 30) does not show that SAA3 is knocked out from WAT, or that any SAA subtypes are knocked out from liver.

We thank the reviewer for pointing out our oversight. The mouse model used in reference 30 of the old manuscript has the same genotype as that of the mice used in this study. However, the paper does not show the absence of SAA3 or other subtypes in adipose tissues. Only lack of expression of the isoforms in the plasma, colon and livers were shown. As explained above, we have now included new figures (2B, 2C, S1B and S1D) to demonstrate the validity of the model.

Also, the other reference they listed for the TKO mouse model (28) is incorrect.

We thank the reviewer for pointing out the mistake, we have now included the correct references in the revised manuscript. 

Measurement of plasma SAA levels also should be provided for the TKO mice with an without the HFHSC diet as further model validation.

In response to reviewer’s comment, we have included a new figure 2D, which shows results from western blot analysis of plasma from experimental mice and demonstrates lack of detectable SAA protein in the TKO mice in the revised manuscript. The antibodies used recognizes all three isoforms of SAA, SAA1,2 and 3. We mixed two different antibodies which were validated (Tannock et al., 2018; J Lipid Res, 59:339) to specifically recognize SAA1,2 and SAA3 respectively. 

We have included the details of genomic organization of the SAA1.1 and SAA2.1 genes and the construction of SAA1.1/2.1-deficient (SAAKO) mice as described in our earlier paper (de Beer et al., J Lipid Res. 2010. 51:3117-3125). TKO mice are constructed by inserting a stop codon into exon 2 of the SAA3 locus in SAAKO mice by CRISPR- CAS9 technology. The method of construction of TKO mice from SAAKO mice is described earlier by Lee et al., 2020 (Lee et al., Cell 2020. 180:79-91 e16; genotyping for SAA3 deficiency is provided in S1 Figure). We have included the details in the revised manuscript in the ‘Materials and Methods’ section, under ‘Animals’ (Highlighted text). We have included a new figure supplement (S1 supplement) showing the PCRs and the banding patterns performed to genotype TKO mice and added the information in the revised manuscript in the ‘Materials and Methods’ section, under ‘Animals’ (Highlighted text). 

The age at which the mice were started on the high fat diet should be stated.

We thank the reviewer for pointing out the mistake of not including the age of the mice at the start of the experiment, we have now included the data in the “Materials and Methods” under the heading “Diet and study design” (highlighted text). 

The authors should quantify adipocyte size. Based on the images presented in Fig. 7, it looks as if the HFHSC-fed TKO mice had larger adipocytes.

In response to the reviewer’s comment, we have now included new data showing mean adipocyte area for WT and TKO mice fed either chow or HFHSC diet (new figure Fig.7B). There was a trend for increased adipocyte size in TKO male mice fed the HFHSC diet, however the difference was not significant when compared to the WT male mice fed the same diet. 

The authors only included 5 mice per group. This seems very small - was a power calculation performed to determine that 5 mice/group would be sufficient?

We included both male and female mice in the study and have chosen to present the data in sexes separately. Due to normal variations in breeding, we had higher numbers of female than male mice. However, for most data the effect of diet and genotype were the same or similar in both sexes, providing further support of the findings. Thus, while some data shows only 5 mice per group per sex, collectively there are sufficient numbers of mice to meet or exceed the numbers indicated from the power calculations performed prior to study initiation. Not all analyses or measures were performed on every mouse, and each figure legend specifies the number of mice included in that figure. 

The discussion section is too short and should be expanded to include the issues brought up in the general section of this review.

In response to the reviewer’s comment, we have now expanded the discussion section to include the issues raised in general about the topic.

---

## [Decision Letter · Decision Letter 1]

9 Feb 2022

PONE-D-21-32072R1Serum Amyloid A is not obligatory for high-fat, high-sucrose, cholesterol-fed diet-induced obesity and its metabolic and inflammatory complicationsPLOS ONE

Dear Dr. Shridas,

Thank you for submitting your manuscript to PLOS ONE. After careful consideration, we feel that it has merit but does not fully meet PLOS ONE’s publication criteria as it currently stands. Therefore, we invite you to submit a revised version of the manuscript that addresses the points still raised by two of the reviewers.

We look forward to receiving your revised manuscript.

Kind regards,

Michael Bader

Academic Editor

PLOS ONE

Journal Requirements:

Reviewers' comments:

Reviewer's Responses to Questions

**Comments to the Author**

1. If the authors have adequately addressed your comments raised in a previous round of review and you feel that this manuscript is now acceptable for publication, you may indicate that here to bypass the “Comments to the Author” section, enter your conflict of interest statement in the “Confidential to Editor” section, and submit your "Accept" recommendation.

Reviewer #1: (No Response)

Reviewer #3: All comments have been addressed

2. Is the manuscript technically sound, and do the data support the conclusions?

Reviewer #1: Partly

Reviewer #3: Yes

3. Has the statistical analysis been performed appropriately and rigorously? 

Reviewer #1: Yes

Reviewer #3: Yes

4. Have the authors made all data underlying the findings in their manuscript fully available?

Reviewer #1: Yes

Reviewer #3: Yes

5. Is the manuscript presented in an intelligible fashion and written in standard English?

Reviewer #1: Yes

Reviewer #3: Yes

6. Review Comments to the Author

Reviewer #1: The authors have provided responses and revisions to the manuscript that substantially improve the work. Nevertheless, the phenotype of the triple-SAA-knockout mice is still lacking robust endpoints. Specifically, the western blot raw data provided implicate that the only antibody used for the assessment of the SAA proteins (Figure 2B&D and supplemental material) was one from Abcam, which recognizes SAA1/2. Despite it being written in the results section, there is no indication that an SAA3-specific primary antibody was used. The rabbit anti-mouse SAA3 antibody from Dr. Scherer is going on 20 years old at this point, and it hasn't been available from him for a decade. It is not convincing that the results presented in the revised manuscript demonstrate an absence of SAA3 in these triple-SAA-knockout mice. The anti-SAA3 antibody should be used alone in a similar western blot. Alternatively, high-quality and specific SAA3 ELISAs are commercially available, and should be used to finally document the phenotype of these triple knockout mice as a complement to the SAA1/2 analysis already included.

Reviewer #3: The revised manuscript is much improved from the earlier version and is responsive to the suggestions from the reviewers. In particular, the abstract now describes the main findings of the study and makes an appropriate conclusion.

Two minor points should be addressed:

(1) The authors have now measured adipocyte size in the wild type and triple knockout mice and claim that although there is a trend towards larger fat cell size in the knockout group (line 296), the differences were not statistically different. However, this conclusion is based on 3 views from only 2 mice in each group. Clearly, a greater samples size is required for adequate comparison between groups.

(2) Line 275: In contrast to the study cited by Ather and Poynter that showed weight gain in Saa3 deficient mice, a study by den Hartigh et al showed that SAA3 deficient mice actually lost weight in response to an obesogenic diet (PMID: 25251243). This reference should also be included in this paragraph, even though they both are cited in the discussion section.

7. PLOS authors have the option to publish the peer review history of their article (what does this mean?). If published, this will include your full peer review and any attached files.

Reviewer #1: No

Reviewer #3: No

---

## [Author Response · Author response to Decision Letter 1]

23 Mar 2022

Response to Reviewer’s comment:

We thank the reviewers for their positive comments recognizing that the manuscript entitled “Serum Amyloid A is not obligatory for high-fat, high-sucrose, cholesterol-fed diet-induced obesity and its metabolic and inflammatory complications” (PONE-D-21-32072) has merit to be published in PLOS ONE. As requested, we provide a point-to-point response to all reviewer’s comments below. Changes in the resubmitted manuscript are indicated as yellow highlighted text (please see ’Revised manuscript with Track Changes’).

In the revised manuscript, we have added one more author, Avery C Patrick, who carried out some of the experiments suggested by the reviewers. 

New and revised figures are described in our point-to-point response, below. 

Response to Associate Editor’s comment:

1.Please review your reference list to ensure that it is complete and correct. If you have cited papers that have been retracted, please include the rationale for doing so in the manuscript text, or remove these references and replace them with relevant current references. Any changes to the reference list should be mentioned in the rebuttal letter that accompanies your revised manuscript. If you need to cite a retracted article, indicate the article’s retracted status in the References list and also include a citation and full reference for the retraction notice.

In response to the academic editor’s advice, we have reviewed the reference list and made sure that it is complete and correct. The list does not contain any retracted references. 

Response to Reviewer’s comment

Reviewer 1:

Reviewer #1: The authors have provided responses and revisions to the manuscript that substantially improve the work. Nevertheless, the phenotype of the triple-SAA-knockout mice is still lacking robust endpoints. Specifically, the western blot raw data provided implicate that the only antibody used for the assessment of the SAA proteins (Figure 2B&D and supplemental material) was one from Abcam, which recognizes SAA1/2. Despite it being written in the results section, there is no indication that an SAA3-specific primary antibody was used. The rabbit anti-mouse SAA3 antibody from Dr. Scherer is going on 20 years old at this point, and it hasn't been available from him for a decade. It is not convincing that the results presented in the revised manuscript demonstrate an absence of SAA3 in these triple-SAA-knockout mice. The anti-SAA3 antibody should be used alone in a similar western blot. Alternatively, high-quality and specific SAA3 ELISAs are commercially available, and should be used to finally document the phenotype of these triple knockout mice as a complement to the SAA1/2 analysis already included. 

We understand the concerns of the reviewer. In response to the concern, we have now included a new data showing the complete lack of SAA3 in the plasma of the triple-SAA-knockout mice (new Fig. 2F; results in page number 9, 2nd paragraph, lines 9-11). The assay was performed with a SAA3-specific elisa kit, ZMSAA3-12K; EMD Millipore, Danvers, MA; the specificity of this elisa kit has been previously tested and reported; Tannock et al., J. Lipid Res. 2018. 59: 339–347. In addition, we have the data showing the absence of SAA3 mRNA in the adipose tissues as well as livers of TKO mice used in the experiment (S1B and S1D Figures respectively). 

Reviewer #3: The revised manuscript is much improved from the earlier version and is responsive to the suggestions from the reviewers. In particular, the abstract now describes the main findings of the study and makes an appropriate conclusion.

Two minor points should be addressed:

(1) The authors have now measured adipocyte size in the wild type and triple knockout mice and claim that although there is a trend towards larger fat cell size in the knockout group (line 296), the differences were not statistically different. However, this conclusion is based on 3 views from only 2 mice in each group. Clearly, a greater samples size is required for adequate comparison between groups.

In response to the comment, we now have adipocyte size measurements from 3 views and 3 mice in each group. We conclude that there are no significant strain-dependent differences in adipocyte sizes with either diet and sexes (revised figure 7B). The method is revised in the ‘Materials and Methods’, under the sub-heading ‘Histology’, page number 7 and described in the ‘Results’ section, page number 13 (highlighted text). 

(2) Line 275: In contrast to the study cited by Ather and Poynter that showed weight gain in Saa3 deficient mice, a study by den Hartigh et al showed that SAA3 deficient mice actually lost weight in response to an obesogenic diet (PMID: 25251243). This reference should also be included in this paragraph, even though they both are cited in the discussion section.

As suggested by the reviewer, we have now included the study results from de Hartigh et al in the results section of the revised manuscript, page number 12-13, lines 281-283.

---

## [Editor Report · Decision Letter 2]

25 Mar 2022

Serum Amyloid A is not obligatory for high-fat, high-sucrose, cholesterol-fed diet-induced obesity and its metabolic and inflammatory complications

PONE-D-21-32072R2

Dear Dr. Shridas,

We’re pleased to inform you that your manuscript has been judged scientifically suitable for publication and will be formally accepted for publication once it meets all outstanding technical requirements.

Kind regards,

Michael Bader

Academic Editor

PLOS ONE
---

## [Editor Report · Acceptance letter]

7 Apr 2022

PONE-D-21-32072R2 

Serum Amyloid A is not obligatory for high-fat, high-sucrose, cholesterol-fed diet-induced obesity and its metabolic and inflammatory complications 

Dear Dr. Shridas:

I'm pleased to inform you that your manuscript has been deemed suitable for publication in PLOS ONE. Congratulations! Your manuscript is now with our production department. 

Kind regards, 

on behalf of

Prof. Michael Bader 

Academic Editor

PLOS ONE